# Emergency MRI in Spine Trauma of Children and Adolescents—A Pictorial Review

**DOI:** 10.3390/children10071094

**Published:** 2023-06-21

**Authors:** Aapo Sirén, Mikko Nyman, Johanna Syvänen, Kimmo Mattila, Jussi Hirvonen

**Affiliations:** 1Department of Radiology, University of Turku and Turku University Hospital, Kiinamyllynkatu 4-8, 20520 Turku, Finland; mikko.nyman@tyks.fi (M.N.); kimmo.mattila@tyks.fi (K.M.); jussi.hirvonen@utu.fi (J.H.); 2Department of Pediatric Orthopedic Surgery, University of Turku and Turku University Hospital, 20520 Turku, Finland; johanna.syvanen@tyks.fi; 3Medical Imaging Center, Department of Radiology, Tampere University and Tampere University Hospital, 33100 Tampere, Finland

**Keywords:** magnetic resonance imaging, pediatric, trauma, spine, emergency imaging

## Abstract

Severe spinal trauma is uncommon in the pediatric population, but due to the potentially devastating consequences of missed injury, it poses a diagnostic challenge in emergency departments. Diagnostic imaging is often needed to exclude or confirm the injury and to assess its extent. Magnetic resonance imaging (MRI) offers an excellent view of both bony and soft tissue structures and their traumatic findings without exposing children to ionizing radiation. Our pictorial review aims to demonstrate the typical traumatic findings, physiological phenomena, and potential pitfalls of emergency MRI in the trauma of the growing spine.

## 1. Introduction

Imaging plays a crucial role in the workup of suspected pediatric spinal trauma [1]. The accuracy of clinical decision tools in excluding cervical spine injury is limited in the pediatric population, especially in children under nine years of age [2]. There are no reliable clinical decision-making rules for thoracolumbar injuries either. In clinical work, the selection of a suitable imaging modality is considered to involve a trade-off, not only between usability and accuracy but also in detecting fractures and soft tissue injuries. Plain radiographs and computed tomography (CT) are the basis of the diagnostic workup unless neurological symptoms occur [1,3]. Mounting evidence demonstrates MRI to be highly applicable in clearing the pediatric spine [4,5,6,7]. MRI seems to be highly accurate not only in ligament tears and other soft tissue injuries but in fractures too [8]. MRI has been shown to be 100% sensitive in unstable cervical spine injuries [9]. The lax ligaments and incomplete ossification make the pediatric population prone to ligamentous injuries. The relatively large head predisposes younger children to a craniocervical junction and upper cervical spine injuries. After nine years of age, the injury profile becomes more comparable to that of adults—fewer injuries in the upper cervical spine and more in the subaxial cervical spine [10]. Spinal cord injury without radiographic abnormality (SCIWORA) is an injury almost exclusively seen in children [11,12]. In SCIWORA, MRI is the only imaging modality able to demonstrate the extent of spinal cord injury [12,13]. Overall, MRI offers superior sensitivity and good specificity in all types of pediatric spinal trauma. The safety profile of emergency MRI is excellent [9], and, most importantly, MRI does not expose children to ionizing radiation.

MRI is more expensive and time-consuming per single scan than plain radiographs and CT. Its availability is often limited, and the drawbacks also include a more frequent need for anesthesia to obtain diagnostic image quality, especially in the preschool-aged population. However, in addition to the health benefits of accurate imaging, the overall cost-effectiveness of emergency MRI in clearing the cervical spine of an obtunded pediatric patient has been found to be positive due to shorter ICU and hospital stays [14]. With skilled MRI technicians, a peaceful and child-friendly environment, and parents’ support, most children aged five years or more can be scanned without anesthesia or light sedation [15,16].

Our pictorial review provides an overview of certain traumatic findings and some physiological phenomena in emergency MRI of the pediatric spine (Table 1). The majority of the knowledge about spinal trauma imaging is based on studies with adult subjects. When applicable, we have focused on papers with children as the scope. The unique features of non-accidental trauma are not discussed.

## 2. Suggested Imaging Protocols and Practical Considerations

According to the current ACR Appropriateness Criteria on Suspected Spine Trauma in children, MRI is categorized as “May Be Appropriate” regarding its consideration as an alternative to plain radiographs for initial imaging in children under 16 years of age with a suspected cervical or thoracolumbar spine injury and the risk factors justifying spinal imaging [1]. MRI is indicated in suspected spinal cord or craniocervical injuries [3] and is helpful in assessing any suspected spinal injury, either for initial imaging or confirmatory imaging after radiographs or CT. Especially if a ligamentous injury is suspected, MRI is useful. Considering osseous structures, MRI is useful in differentiating acute injuries with bone marrow edema from chronic injuries and anatomical variants. MRI is also able to demonstrate bony injuries without fracture lines or other structural changes visible in radiographs or CT.

Younger (particularly <5 years of age) and critically ill children require sedation through anesthesia to achieve sufficient imaging quality. Careful decision making regarding the most suitable imaging method must be conducted case by case in this age group. The first principle should be adequate imaging for prompt diagnosis. Nonetheless, the risks of ionizing radiation and sedation must also be taken into consideration when planning the imaging of these vulnerable patients [17]. Runge et al. showed that in children aged 4–6 years, the need for anesthesia could be reduced with the use of interactive mobile phone applications and toy scanners before the actual scanning, a child-friendly multimedia environment in the scanning room, and staff dedicated to pediatric imaging [15]. Mobile applications or toy scanners are often not applicable in the case of acute trauma, but the general methods of working with pediatric patients (patience, flexibility, containment of emotions) [16] will decrease the need for anesthesia also in trauma patients. In our clinical practice, we have successfully scanned many children aged 4–6 years, with some additional time used to ease the patient’s anxiety in cooperation with the child’s parents. We have found the multimedia environment [15] useful too. Children aged 6 years or more can usually be scanned without anesthesia.

Any patient with high-energy trauma should be examined according to standardized trauma protocols, including whole-body and cervical CT scans [18], followed by MRI if needed. On the other hand, CT can be used for complementary imaging if the MRI leaves any doubt about the morphology of an osseous injury, as radiologists are usually more experienced in assessing fractures with CT. Targeted, small-FOV CT of the fracture site is usually sufficient to achieve the needed information. However, our experience is that CT seldom delivers additional information of clinical relevance if the MRI is technically adequate. The use of zero echo time (ZTE) sequences with a high spatial resolution in osseous structures can further reduce the need for CT in bony injuries [19,20]. Considering technical innovations, any means to speed up the MRI image acquisition are helpful in emergency imaging, especially in the pediatric population [21].

Our routine imaging protocol includes sagittal T1- and T2-weighted, sagittal and coronal STIR, and axial T2-weighted. In the case of either clinical or radiological suspicion of a ligamentous injury in the upper cervical spine and craniocervical junction, the following series with a small field of view (FOV) are obtained: sagittal and coronal PD-weighted and axial T2-weighted. Contrast media are not used in the routine protocol. The device used is a Philips Ingenia 3 Tesla scanner with a Philips dStream coil system (Philips Healthcare, Best, The Netherlands). Detailed specifications of the routine sequences adopted in our clinical practice can be found in Appendix A.

In concordance with previously published observations [22], we have found the STIR sequence to be very important due to its high sensitivity in injury detection. T1- and T2-weighted imaging are needed to assess the structural injuries further. T2-weighted imaging is also crucial in examining the spinal cord.

## 3. Fractures

MRI has a high negative predictive value in spinal fractures. Fat-suppressed sequences, particularly short tau inversion recovery (STIR), are the most sensitive in acute trauma [23]. T1- and T2-weighted sequences help to assess the exact fracture morphology. Novel high-resolution sequences optimized for bony structures improve the accuracy of MRI in fractures. Children are prone to multiple spinal fractures, often at non-contiguous levels. Hence, the inspection must be careful and thorough, even if—and especially if—one fracture is found [24,25].

### 3.1. Compression Fractures

Compression fractures may occur in any spinal segment. Over 80% of these injuries in children affect more than one level [26,27]. Compressions are often non-junctional [26]. MRI displays compression fractures and vertebral body contusions even without visible structural height loss (Figure 1). In burst fractures, MRI enables an instant assessment of concurrent medullary injuries and spinal canal hematomas (Figure 2). Usually, burst fractures are primarily seen in whole-body trauma CT performed due to high-energy injury mechanisms.

### 3.2. Chance Fractures

Pediatric chance fractures are rare high-energy injuries often caused by motor vehicle accidents. The most commonly affected levels are L2 and L3 [19]. CT is crucial for the initial assessment of a potential polytrauma patient. As secondary imaging, MRI can distinguish physeal and discal involvement, ligamentous injuries, and spinal cord injuries in addition to the course of the fracture lines [28,29,30,31] (Figure 3).

### 3.3. Avulsion Fractures

Most avulsion fractures of the pediatric spine occur in levels C0–C2, being avulsions of craniocervical ligaments, the alar ligament in particular [32,33,34]. The alar ligament may avulse from its origo in the occipital condyle (Figure 4) or its insertion in the dens.

Acute clay-shoveler-type spinous process avulsion fractures are also seen, usually in teenage athletes [35]. On MRI, chronic stress fractures of the spinous process (sometimes called apophysitis of the spinous process) [36,37] may have a similar appearance to acute fractures. Chronic spinous process injuries are most abundant in the same type of population as acute fractures, but the symptoms usually follow a more gradual course. Targeted CT of the edematous spinous process helps to separate acute, sharp avulsion fractures from chronic injuries with sclerotic margins and fragmentation (Figure 5).

### 3.4. Other Fractures

Fractures of the vertebral arch, spinous process, transverse process, and facet joints can also be identified in MRI (Figure 6 and Figure 7). As with other fractures, fat-suppressed T2-weighted images are key in evaluating acute post-traumatic bone marrow edema, which may be more difficult to detect in smaller bony structures. Any spinal fracture in children indicates significant trauma energy, underlining the importance of carefully excluding any other traumatic findings.

## 4. Ligamentous Injuries

In ligamentous injuries, MRI is superior to any other imaging modality. Ligamentous discontinuity and possible spinal malalignment are most easily assessed on T2-weighted and STIR sequences. STIR demonstrates an edematous signal in partial tears or ligament strains without visible structural changes. The threshold for performing MRI should be especially low in suspected craniocervical junction trauma [3].

### 4.1. Occipitocervical and Atlantoaxial Ligaments

The craniocervical and atlantoaxial joints are mobile structures allowing flexion–extension movement and rotation of the head. The joints’ biomechanics and the importance of different stabilizing structures are not yet completely understood. The joint capsules, the alar ligaments, and the transverse ligaments are crucial to stability [33,38,39]. The role of the tectorial membrane is more controversial, but it might also be of importance in preventing overextension [33,40]. Injury to the stabilizing joints or ligaments may lead to instability. Isolated soft tissue injuries and avulsion fractures without complete joint dissociation (Figure 4) may occur. All the joints and ligaments mentioned above are visible on MRI. A higher field strength and dedicated proton density- and T2-weighted sequences with a smaller field of view may help to delineate these small structures better [41]. Figure 8 demonstrates a case with an upper cervical spine ligament injury.

### 4.2. Posterior Ligament Complex

The posterior ligament complex (PLC) consists of the ligamentum flavum, interspinous ligament, supraspinous ligament, and facet joint capsules. It is noted as an essential factor for spinal stability in numerous classification systems proposed for spinal trauma [42]. Of these classifications, the TLICS [43,44,45] and AO Spine systems [43,46,47,48] have also been found to have good interrater reliability in the pediatric population. Dawkins et al. [44], however, concluded the interrater reliability of the TLICS classification to be lower with patients having undergone MRI than those treated based on CT imaging only. As the authors discussed, this is probably explained by MRI’s superior sensitivity in demonstrating stable PLC injuries that would be undetectable in CT or plain radiographs. It is possible that the MRI’s suggested poor interrater reliability with the TLICS classification could be improved with education, given that, in the study by Dawkins et al., spine surgeons with varying experience read the MRIs, and no radiologists were involved.

In addition to a widened interspinous distance and other indirect measures used with CT and plain radiographs, MRI can differentiate the very components of the PLC, revealing the culprits responsible for the indirect CT findings (Figure 9).

### 4.3. Anterior and Posterior Longitudinal Ligaments

Anterior and posterior longitudinal (ALL and PLL) injuries occur as part of a gross fracture–dislocation. The literature on the pediatric population’s isolated or non-dislocated ALL or PLL injuries is scarce. These injuries might be rare, especially in younger children [49], but the overall incidence is unknown. Moreover, in our clinical practice, we have found non-dislocated ALL and PLL injuries uncommon. An example of a discrete, partial ALL tear as a part of a non-dislocated flexion–extension cervical spine injury is presented in Figure 10.

## 5. Other Soft Tissue Injuries

### 5.1. Intervertebral Disc Injuries

As with any intervertebral disc pathology, acute traumatic disc injuries are best seen with MRI [50]. The high signal from post-traumatic edema can best be separated from the high physiological signal originating from the nucleus pulposus using fat-suppressed T2-weighted images. Degenerative intervertebral disc changes are not rare in the pediatric population [51,52,53], and these should not be confused with acute discal injury with the edematous signal. Figure 11 demonstrates a solitary acute intervertebral disc injury.

### 5.2. Muscle Injuries

MRI is the gold standard in the imaging of muscular trauma [54]. Spinal muscle trauma seldom needs any specific treatment. Muscle trauma can best be appreciated on fat-suppressed T2-weighted images, where it appears as areas of high signal (post-traumatic edema) (Figure 12). The hematomas associated with major muscular injuries are also visible in fat-suppressed T2-weighted images.

### 5.3. Nerve Plexus Injuries

Treatment planning in nerve plexus injuries requires dedicated imaging [55,56]. However, in a trauma patient with neurological symptoms in the extremities, the emergency spinal MRI can be extended to immediately confirm or exclude a possible nerve plexus injury (Figure 13). If the dedicated neurography sequences are unavailable, routine fat-suppressed T2-weighted imaging provides good sensitivity in plexus injury detection [56].

### 5.4. Vascular Injuries

CTA has been a workhorse in suspected vascular trauma for decades. On a routine cervical spine MRI, an arterial injury can be detected as an abnormal flow void in T1-weighted, T2-weighted, STIR, and proton density sequences (Figure 14). Vessel wall injury (dissection or intramural hematoma) can be seen in routine sequences. When in doubt, modern vessel wall imaging techniques with a high spatial and contrast resolution can significantly improve the detection and characterization of even minor arterial injuries [57,58,59].

## 6. Spinal Cord Injuries

The use of MRI in acute spinal cord trauma was first described in 1983 [60,61] and it has been the gold standard in imaging these injuries ever since. T1-weighted, T2-weighted, and STIR sequences are the cornerstones in assessing the cord [49]. Diffusion-weighted imaging (DWI) and diffusion tensor imaging (DTI) might be useful in detecting subtle injuries and as a prognostic biomarker, but their role in clinical practice is not fully established yet [62,63,64,65,66]. Susceptibility-weighted imaging (SWI) or T2*-weighted sequences might increase the sensitivity in the case of small intramedullary hemorrhages but are technically challenging to obtain due to pulsation and motion artifacts [67,68]. Cases of spinal cord injuries are presented in Figure 15 and Figure 16.

Spinal cord injury without radiographic abnormality (SCIWORA) is mainly seen in children, although it also exists in adults. The term was first described by Pang and Wilberger in 1982 [11], but symptomatic spinal cord injury without spinal fracture or dislocation was recognized even earlier [69]. SCIWORA is thought to occur due to the significant mobility and laxity of the spine in children, allowing self-reducing displacement to damage the cord [12]. The introduction and development of MRI have led to a terminological discussion evolving the concept of SCIWORA [13,70] by providing visibility to the cord itself and sometimes revealing structural injuries not seen in radiographs or CT [71]. Nevertheless, there seems to be a small number of patients with neurological symptoms from cord injury not visible in contemporary MR imaging, eventually with a favorable long-term outcome [12,72].

## 7. Physiological Findings and Pitfalls

There are many practical issues to take into account when interpreting the emergency MRI of the pediatric spine. Some of these are universally related to MRI, but children have many peculiarities that the radiologist and the physician must be aware of. The following list is not comprehensive but might help in avoiding some pitfalls.

### 7.1. Pseudosubluxation of the Cervical Spine

Pseudosubluxation in the subaxial cervical spine is a normal variant in younger children, and it is essential to distinguish it from true subluxation. Pseudosubluxation is most commonly seen in level C2/3, followed by level C3/4 [73,74,75,76]. On MRI, the lack of bony or soft tissue edema in addition to the normal posterior cervical line (Swischuk line) [74] makes the recognition of pseudosubluxation more straightforward than with the other imaging modalities (Figure 17).

### 7.2. Vertebral Wedging

Anterior wedging of the vertebral bodies is a normal physiological phenomenon that disappears gradually with skeletal maturation [77,78,79]. With MRI, physiological wedging is readily distinguished from compression fractures by the lack of bone marrow edema or a visible fracture line (Figure 18).

### 7.3. Juvenile Spondylolysis

Juvenile lumbar spondylolysis ensues from repetitive stress, but the onset of symptoms can be sudden, and the chronic pain may be exaggerated by acute trauma. The clinical presentation and the typical location of the findings centered in the pars interarticularis help to distinguish a stress injury from an acute traumatic fracture. The spondylolysis and surrounding bone marrow edema can probably be assessed with MRI [80], but the evidence of MRI’s sensitivity is not fully concurrent [81]. However, it seems that MRI’s performance can be improved with high-resolution T1-weighted sequences optimized for bony structures [82] or with a novel ultrashort time-to-echo technique [83]. An example of a lumbar spondylolysis is presented in Figure 19. Targeted small-FOV CT may be used in estimating the age, grade, and bony union rate of the spondylolysis in treatment planning and follow-up [82,83,84,85,86].

### 7.4. Imaging Appearances of Normal Skeletal Maturation

Every radiologist performing pediatric imaging must be aware of the fundamentals of skeletal maturation [87]. On the MR imaging of acute pediatric trauma, one of the potential pitfalls is the physiological high T2 signal at the physis and metaphyseal spongiosa of the secondary ossification centers [88]. This can be misinterpreted as traumatic edema if the normal anatomy and development of the ossification centers are not kept in mind (Figure 20). On the other hand, an unfused vertebral ring apophysis [89,90], apophyseal injuries [91], and other calcifications not related to the acute injury are readily distinguished from fractures with the absence of edema (Figure 21).

### 7.5. Cerebrospinal Fluid Pulsation Artifacts

The pulsatile movement of cerebrospinal fluid (CSF) causes flow void artifacts, especially on T2-weighted sequences. The artifacts can occur anywhere in the subarachnoid spaces, but, in the spine, they are often most prominent in the cervical and thoracic regions [92,93]. Usually, the hazy and undelineated appearance of flow void artifacts differentiates them from hematomas or dilated veins. A more conspicuous example of a CSF flow void artifact is annotated in Figure 17, but, to some extent, they can also be seen in Figure 1, Figure 4, Figure 5, Figure 7, Figure 8, Figure 9, Figure 10, Figure 11, Figure 12, Figure 14, Figure 15, Figure 17, Figure 18 and Figure 21 without annotations.

### 7.6. Motion Artifacts

MR imaging is highly susceptible to patient movement. In the pediatric population, motion artifacts may often become an issue. As described above, the circumstances must be appropriate and child-friendly to obtain sufficient image quality. The imaging protocol needs to be optimized to be as quickly as possible, and the most crucial sequences should be obtained first (Figure 22).

### 7.7. Metal-Induced Artifacts

Compared to the adult population, ferromagnetic implants and other foreign bodies in the spinal region are uncommon among children. Dental braces are often seen, but, unlike brain MRI, they seldom cause significant challenges in spinal imaging. Braces inflict susceptibility artifacts, but the extent of the artifact is usually limited outside the area of interest (Figure 23). Naturally, for safety reasons, information about any ferromagnetic foreign body within the patient must be ensured, and the possible effects on MRI safety and image quality must be assessed [94,95].

## 8. Conclusions

MRI offers a radiation-free alternative in imaging pediatric spinal trauma, providing an excellent view for both bony and soft tissue injuries. The need for sedation can be diminished with calm and child-friendly conditions and dedicated staff, especially in children older than 5 years, making MRI more approachable in clinical practice. Younger children require sedation or anesthesia. To understand traumatic findings, the radiologist must be aware of the fundamental characteristics of the growing spine.

## Figures and Tables

**Figure 1 children-10-01094-f001:**
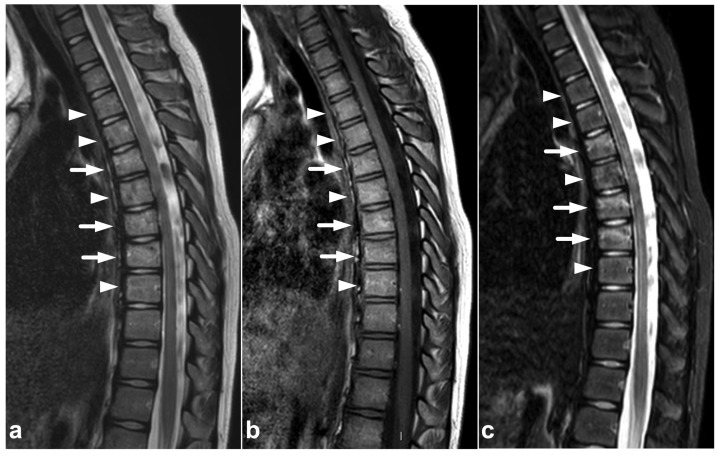
(**a**) Sagittal T2-weighted. (**b**) Sagittal T1-weighted. (**c**) Sagittal STIR. A 9-year-old female, emergency MRI after a horseback riding accident. MRI demonstrates traumatic changes in vertebral bodies Th3–Th9, of which Th5, Th7, and Th8 have structural compressions (arrows). Th3, Th4, Th6, and Th9 have contusions without visible height loss (arrowheads).

**Figure 2 children-10-01094-f002:**
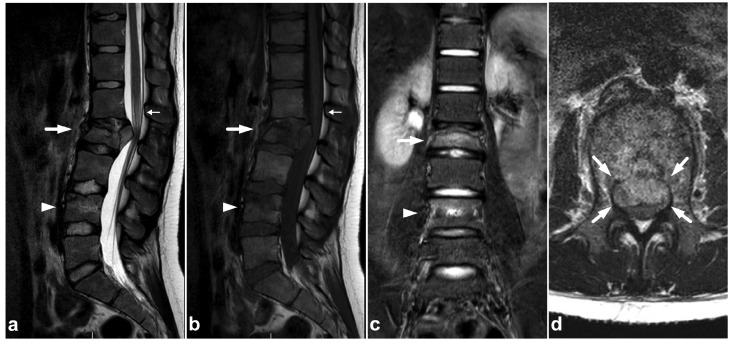
(**a**) Sagittal T2-weighted. (**b**) Sagittal T1-weighted. (**c**) Sagittal STIR. (**d**) Axial T2-weighted. A 16-year-old female, back pain after falling down the stairs. Burst fracture in L2 (arrow) with a fragment protruding into the spinal canal and compressing the conus medullaris. Edema can be seen in the conus medullaris immediately above the most compressed level (small arrow). In L4, there is a stable compression fracture (arrowheads). No intraspinal hematoma can be seen. Minor paraspinal edema/hematoma is best demonstrated in the axial image (**d**).

**Figure 3 children-10-01094-f003:**
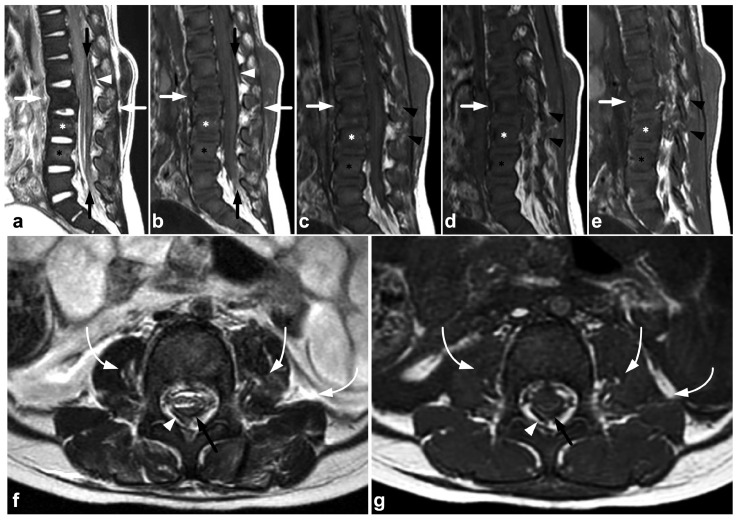
(**a**) Sagittal T2-weighted. (**b**) Sagittal T1-weighted. (**c**) Sagittal left-sided off-midline T1-weighted. (**d**) Sagittal right-sided off-midline T1-weighted. (**e**) Sagittal left-sided off-midline T1-weighted. (**f**) Axial T2-weighted. (**g**) Axial T1-weighted. A 2-year-old female, motor vehicle accident. Chance-type fracture through the vertebral body and the posterior arch is seen in L2 (white arrows). L2/3 and L3/4 facet joints are widened and subluxated (black arrowheads). Kyphosis at level L2. Compression fracture in L3 (white asterisk) and contusion in L4 (black asterisk). Posteriorly, there is a longitudinal intradural hematoma (black arrows) and epidural hematoma (white arrowheads). A paraspinal hematoma (curved arrows) and posterior subcutaneous hematoma are also seen.

**Figure 4 children-10-01094-f004:**
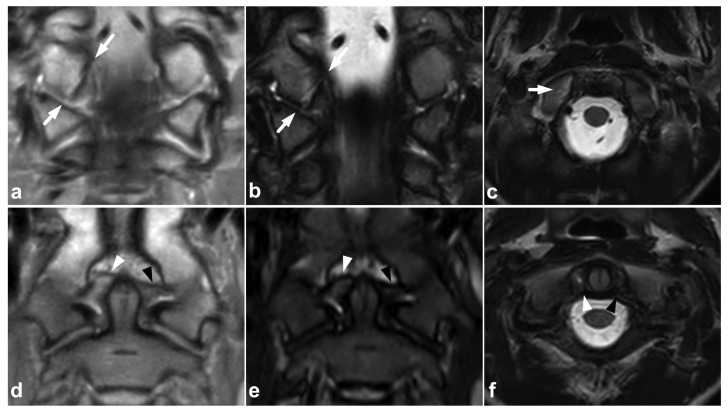
(**a**) Coronal PD-weighted. (**b**) Coronal T2-weighted. (**c**) Axial T2-weighted. (**d**) Coronal PD-weighted. (**e**) Coronal STIR. (**f**) Axial T2-weighted. A 13-year-old male, motor vehicle accident. Avulsion fracture (arrows) of the right alar ligament origo in the occipital condyle, minor dislocation. The right alar ligament (white arrowheads) is swollen and loose but not completely torn. An intact left alar ligament is marked with black arrowheads.

**Figure 5 children-10-01094-f005:**
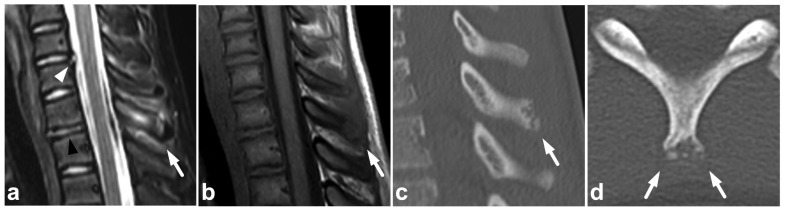
(**a**) Sagittal STIR. (**b**) Sagittal T1-weighted. (**c**) Targeted NECT, sagittal reconstruction. (**d**) Targeted NECT, axial reconstruction. A 13-year-old male with neck pain after an ice hockey accident. MRI reveals significant bone marrow edema in the spinous process of Th1 (arrows). The surrounding soft tissues are also edematous. In targeted CT, the apophyseal region of the spinous process is fragmented without sharply lineated fracture lines. The findings are consistent with a chronic stress injury, perhaps exacerbated by the acute trauma. A dorsal annular tear and a small intervertebral disc protrusion are seen on level C6/7 (white arrowhead). Intervertebral disc Th1–Th2 is dehydrated and lower than adjacent discs (black arrowhead). These findings suggest degenerative changes.

**Figure 6 children-10-01094-f006:**
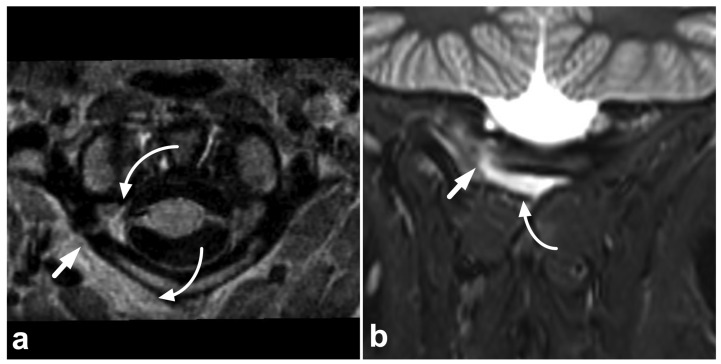
(**a**) Axial PD-weighted. (**b**) Coronal STIR. (**c**) Sagittal T2-weighted. (**d**) Sagittal T1-weighted, off-midline. A 14-year-old female, motor vehicle accident. Non-displaced right-sided fracture at the posterior arch of the C1 is marked with straight arrows. The fracture line extends to the transverse foramen, but the vertebral artery (arrowhead) is intact, with a normal flow void and no intramural hematoma. There is a hemorrhage (curved arrows) around the fracture inside and outside the transverse foramen.

**Figure 7 children-10-01094-f007:**
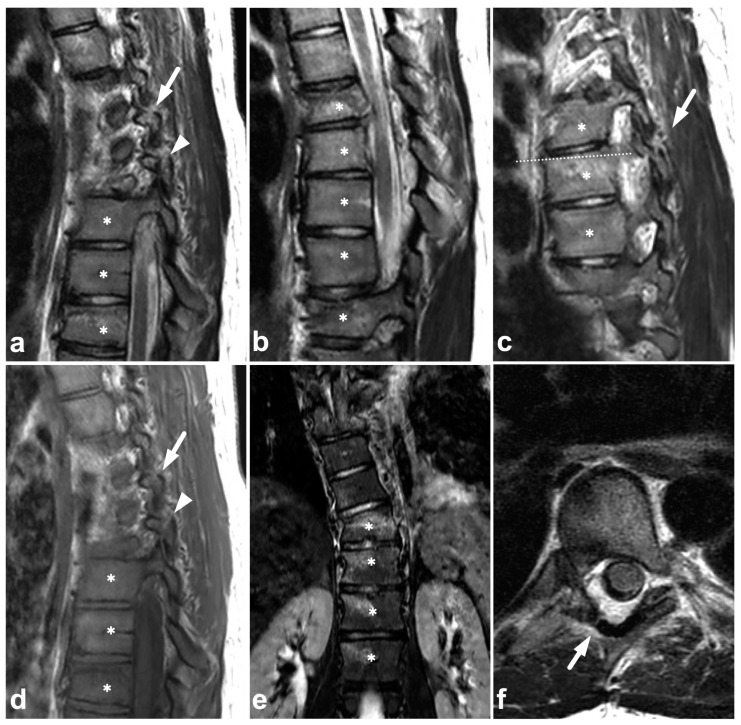
(**a**) Sagittal left-sided off-midline T2-weighted. (**b**) Sagittal T2-weighted. (**c**) Sagittal right-sided off-midline T2-weighted. (**d**) Sagittal left-sided off-midline T1-weighted. (**e**) Coronal STIR. (**f**) Axial T2-weighted (dotted line on the image (**c**)). A 17-year-old female with a history of idiopathic juvenile scoliosis and acute spinal trauma due to a motorcycle accident. The patient had fractures and contusions in multiple vertebrae; the injuries in Th6, Th7, Th8, Th9, and Th10 (asterisks) are seen in the presented images. In addition to vertebral body fractures, the posterior elements were involved bilaterally in Th6 (arrows) and on the left side in Th7 (arrowhead). Facet joints remained congruent. Burst-like morphology is seen on the Th6 fracture. In image (**b**), the fracture seems to dislocate the spinal cord, but in the axial plane (not presented here), the cord was seen not to be compressed.

**Figure 8 children-10-01094-f008:**
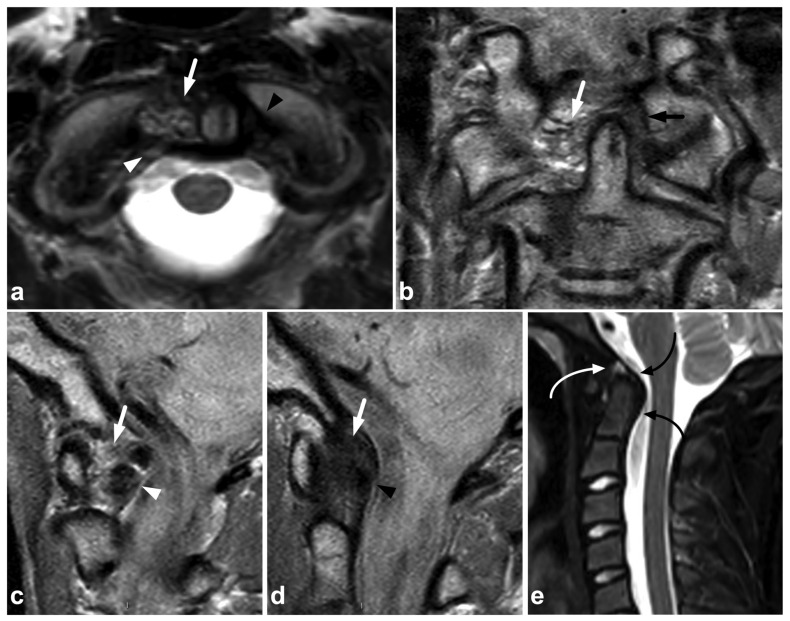
(**a**) Axial T2-weighted. (**b**) Coronal PD-weighted. (**c**) Right-sided off-midline sagittal PD-weighted. (**d**) Sagittal PD-weighted. (**e**) Sagittal STIR. A 12-year-old male, football injury. The right alar ligament (white arrows) is torn. The right side of the transversal ligament is thickened and heterogenous, suggesting a partial distension injury (white arrowheads). The left alar ligament (black arrows) and the central and left-sided portions of the transversal ligament (black arrowheads) are intact. The tectorial membrane is unharmed (curved black arrows), but the apical ligament of the dens is poorly visible, probably torn (curved white arrow). Apparent asymmetry of the lateral atlantodental intervals is seen, but there are no signs of occipitocervical or atlantoaxial joint capsule disruption.

**Figure 9 children-10-01094-f009:**
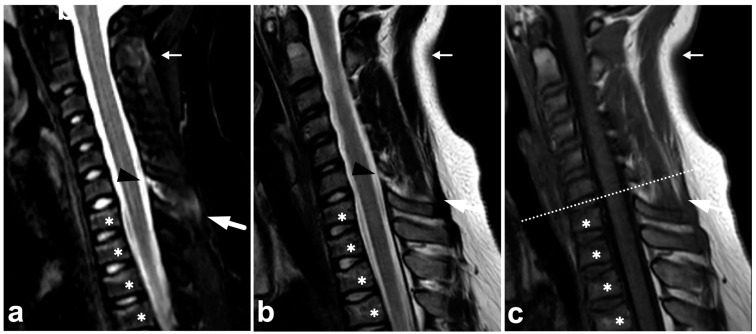
(**a**) Sagittal STIR. (**b**) Sagittal T2-weighted. (**c**) Sagittal T1-weighted. (**d**,**e**) Axial T2-weighted, two adjacent slices at the level C6/7 (line on the image (**c**)). (**f**) Right-sided off-midline sagittal STIR. (**g**) Coronal STIR. (**h**) Left-sided off-midline sagittal STIR. A 9-year-old female, flexion injury in a trampoline accident. On level C2/3, there is slight edema in the posterior atlantoaxial membrane (PAAM), rectus capitis posterior major muscle, nuchal ligament, and adjacent deep cervical fat tissue (small white arrows). On level C6/7, we see more prominent edema of the interspinous and supraspinous ligaments (white arrows); there is a partial tear in the interspinous ligament. At this level, the ligamentum flavum is inhomogeneous (black arrowheads) due to a partial tear but without loss of continuity. A thin hematoma can also be seen between the flavum and posterior arch of C7 (white arrowheads, image (**d**)). Minor facet joint injury is also present; a small amount of fluid and edema can be seen at the left C2/3 facet joint and the posterior parts of both facet joints at level C6/7 (angled arrows). In addition to the PLC injury, there are compression fractures at the anterior parts of the vertebral bodies C7/Th3 (white asterisks). This is also a typical finding in flexion injuries of the cervical spine. The findings presented are highly unlikely to lead to instability, demanding a surgical fixation. However, this case demonstrates MRI’s power to directly assess the different stabilizing structures of the spine.

**Figure 10 children-10-01094-f010:**
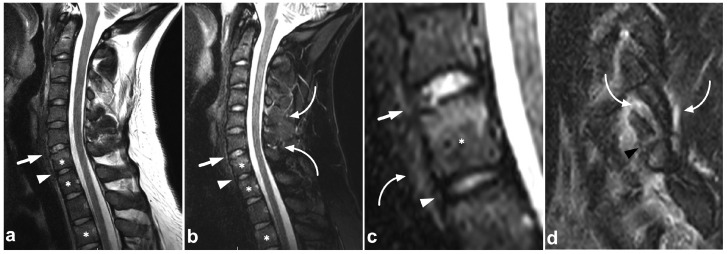
(**a**) Sagittal T2-weighted. (**b**) Sagittal STIR. (**c**) Sagittal STIR, cropped image. (**d**) Sagittal STIR, cropped right-sided off-midline image. A 17-year-old male, motor vehicle accident. Cervical spine injury at level C6/7 in particular. Compression fractures in the vertebral bodies C7, Th1, and Th3 (asterisks) and right-sided processus articular superior fracture in C7 (black arrowhead). ALL is partly torn at level C6/7 (white arrows), best seen in image (**d**). Moreover, the intervertebral disc at level C6/7 is edematous and inhomogeneous due to discal injury. Intact ALL is demonstrated at level C7/Th1 (white arrowhead). Soft tissue edema is seen at the prevertebral space, interspinous ligaments, deep posterior cervical muscles, the interspinous ligament, and inside and outside the facet joint (curved arrows). PLL and ligamentum flavum are intact, and no malalignment is seen.

**Figure 11 children-10-01094-f011:**
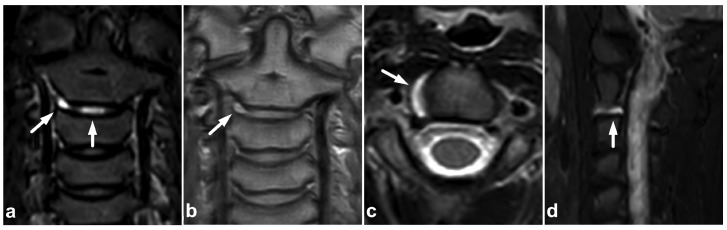
(**a**) Coronal STIR. (**b**) Coronal PD-weighted. (**c**) Axial T2-weighted. (**d**) Sagittal right-sided off-midline STIR. A 10-year-old female, neck pain and severe neck stiffness after a head-first collision when playing. MRI reveals intervertebral disc edema at level C2/3, right-sided annular tear, and lateral disc bulging without neural structure compromise (white arrows). No other signs of injury were seen.

**Figure 12 children-10-01094-f012:**
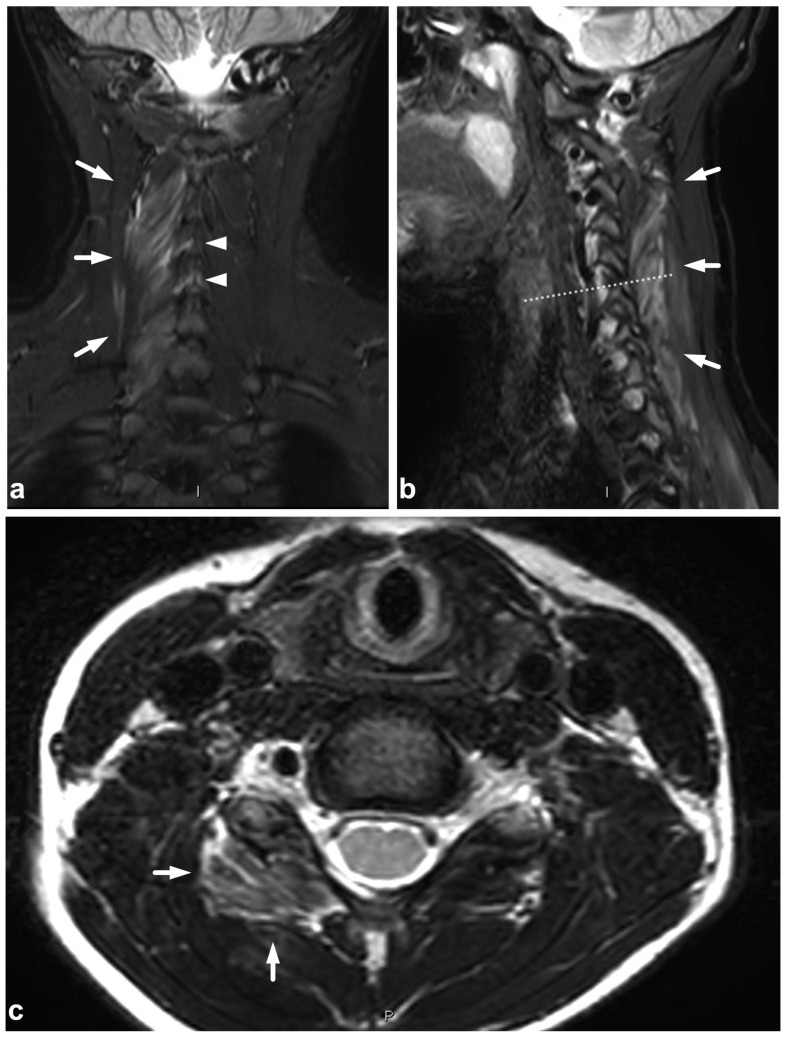
(**a**) Coronal STIR. (**b**) Sagittal STIR, off-midline right-sided image. (**c**) Axial T2-weighted, level C6 (dotted line in the image (**b**)). A 12-year-old male, cervical flexion injury in a trampoline accident. MRI demonstrates right-sided grade 1 muscle injury (white arrows) in the following muscles: rotatores, multifidus cervicis, interspinales, spinalis, and semispinalis. A minor edematous strain of the interspinous ligaments is seen at levels C5/6 and C6/7 (arrowheads).

**Figure 13 children-10-01094-f013:**
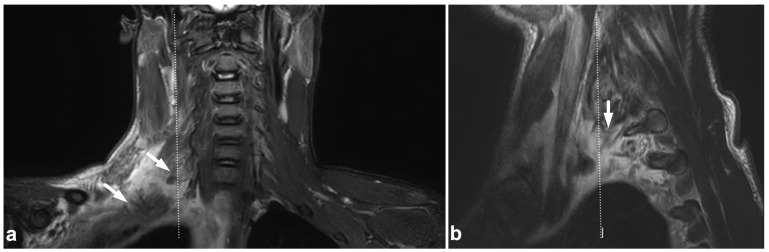
(**a**) Coronal STIR. (**b**) Sagittal T2-weighted. The dotted line in the image (**a**) represents the plane of the sagittal image (**b**) and vice versa. A 13-year-old male after a moped accident. MRI demonstrates a muscle injury and significant edema at the brachial plexus region (arrows). This emergency imaging was not definitive in assessing the extent of the injury. However, extending the cervical imaging to examine the plexus region tentatively helped to confirm the clinical suspicion of upper extremity paralysis being caused by a brachial plexus injury and not a central nervous system injury.

**Figure 14 children-10-01094-f014:**
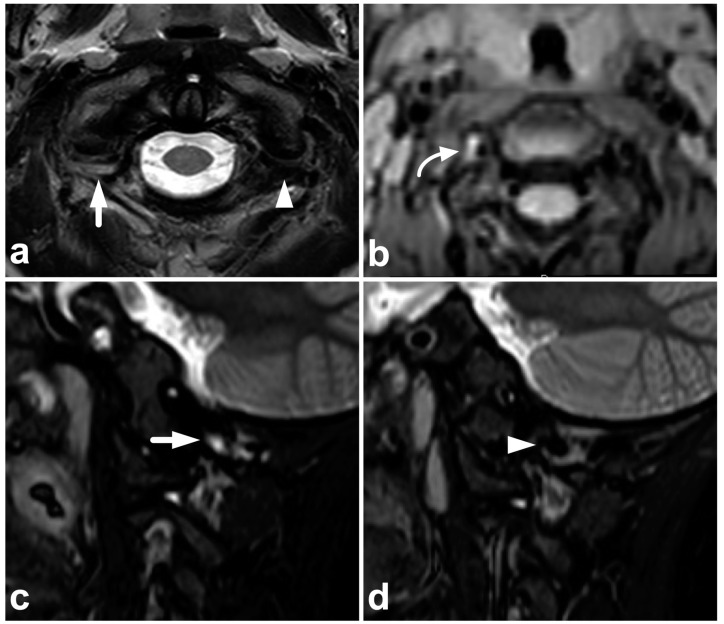
(**a**) Axial T2-weighted. (**b**) Axial plane of an isotropic fat-saturated T1-weighted (black blood sequence). (**c**) Right-sided off-midline sagittal STIR. (**d**) Left-sided off-midline sagittal STIR. (**e**) Axial trace diffusion-weighted image of the brain. (**f**) Axial apparent diffusion coefficient map of the brain. A 5-year-old male, found with impaired consciousness in unclear circumstances. MRI was performed to exclude trauma and revealed an occluding dissection of the right vertebral artery. The occluded right vertebral artery lacked a normal flow void on T2-weighted and STIR images (white arrows), while the patent left vertebral artery presents a normal flow void (white arrowhead). T1 fat-saturated sequence (**b**) reveals an intramural hematoma with a bright T1 signal in the wall of the dissected artery caudally to the completely occluded segment (curved arrow). Diffusion-weighted brain imaging demonstrates right-sided pontine infarction as a sequel to vertebral artery occlusion (black arrowheads). No other findings were suggestive of acute injury in the cervical spine or in the brain. The cause of the dissection remains unknown.

**Figure 15 children-10-01094-f015:**
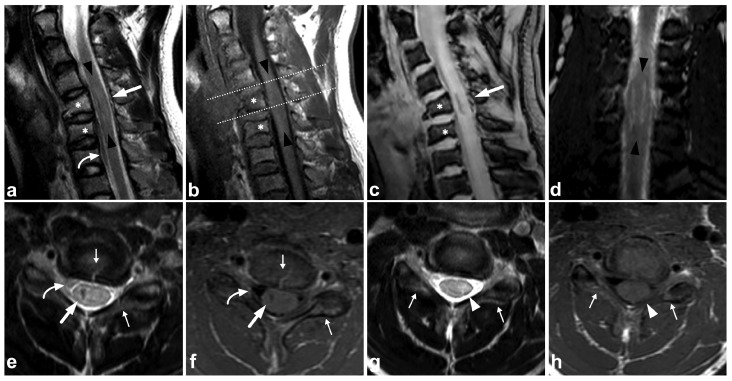
(**a**) Sagittal T2-weighted. (**b**) Sagittal T1-weighted. (**c**) Sagittal T2-weighted fast field echo. (**d**) Coronal STIR. (**e**) Axial T2-weighted, level of the lower end plate of C6 (caudal dotted line in the image (**b**)). (**f**) Axial T1-weighted, level of the lower end plate of C6 (caudal dotted line in the image (**b**)). (**g**) Axial T2-weighted, level of the lower end plate of C5 (cranial dotted line in the image (**b**)). (**h**) Axial T1-weighted, level of the lower end plate of C5 (cranial dotted line in the image (**b**)). A 17-year-old male after diving into shallow water. The spinal cord is edematous approximately from the level of the lower end plate of C4 to the lower end plate of C6 (black arrowheads). There is a hemorrhagic contusion in the right-sided grey matter (white arrows) and a contusion without macroscopic hemorrhage on the left side (white arrowheads). Hematoma can be seen in the anterior epidural space (curved arrows). There are fractures of vertebral bodies and posterior arches of C5 and C6 (asterisks and small arrows).

**Figure 16 children-10-01094-f016:**
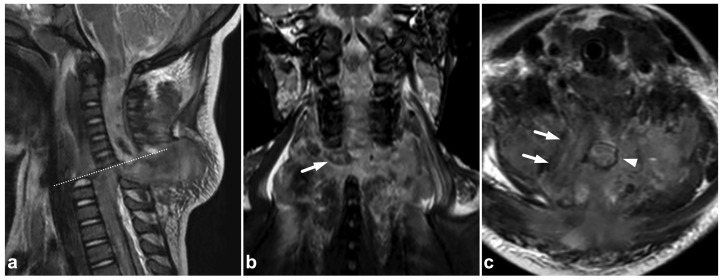
(**a**) Sagittal T2-weighted. (**b**) Coronal STIR. (**c**) Axial T2-weighted (dotted line in the image (**a**)). A 2-year-old female, severe cervical fracture–dislocation after a car accident. The spinal cord is transected. The cord caudally to the transection site (arrows) is dislocated posteriorly and on the right side of the cranial end of the transected cord (arrowhead). An extensive hematoma is seen around the fracture.

**Figure 17 children-10-01094-f017:**
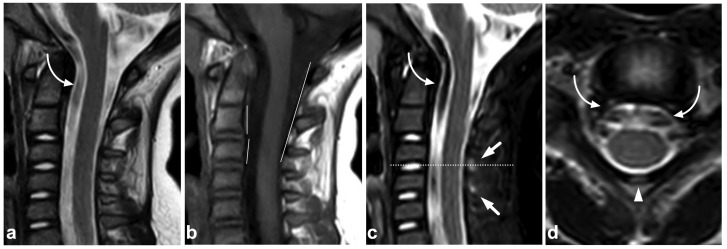
(**a**) Sagittal T2-weighted. (**b**) Sagittal T1-weighted. (**c**) Sagittal STIR. (**d**) Axial T2-weighted (level of the dotted line in the image (**c**)). A 3-year-old male after falling from a height. A cervical spine MRI shows a small malalignment of the vertebral bodies C2 and C3 (short white lines), but the posterior cervical line demonstrates posterior arches to be in line. No edematous changes are seen on this level, confirming the malalignment to be a case of pseudosubluxation. However, there is subtle edema at the anterior part of the interspinous ligament at levels C3/4 and C4/5 (arrows) and a small focus of bright signal in the ligamentum flavum on axial T2-weighted images (arrowhead). These findings indicate a minor, stable PLC injury caudal to the physiological pseudosubluxation. The case also demonstrates the prominent flow void artifacts around the spinal cord (curved arrows) caused by normal liquor pulsation.

**Figure 18 children-10-01094-f018:**
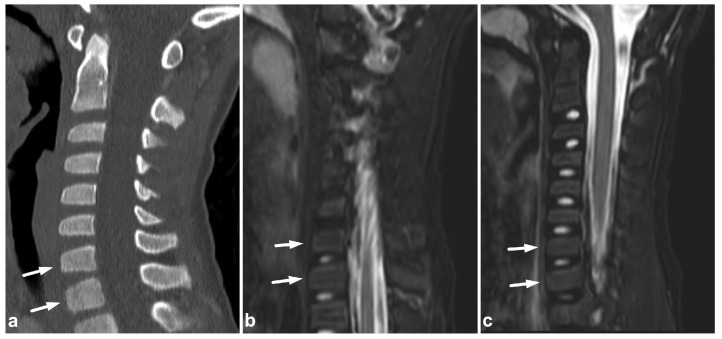
(**a**) Contrast-enhanced trauma CT, sagittal reconstruction with a bone kernel. (**b**) Sagittal STIR. (**c**) Sagittal STIR. A 7-year-old male after falling from a height. Wedge-shaped vertebral bodies seen at the trauma CT, especially C7, and Th1, were ambiguous (arrows). Impaired consciousness hampered the clinical assessment. A single sagittal STIR sequence was obtained to rule out compression fractures, revealing no edema. Some motion artifacts are seen on MRI, as the patient was scanned without sedation. Still, the image quality is sufficient to rule out compression fractures.

**Figure 19 children-10-01094-f019:**
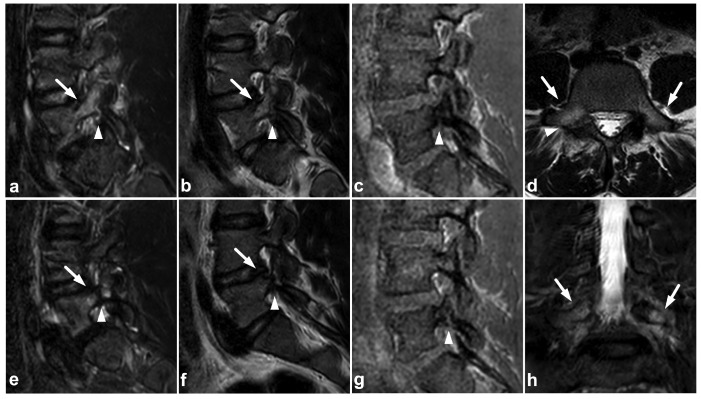
(**a**) Sagittal right-sided off-midline STIR. (**b**) Sagittal right-sided off-midline T2-weighted. (**c**) Sagittal right-sided off-midline black bone sequence. (**d**) Axial T2-weighted. (**e**) Sagittal left-sided off-midline STIR. (**f**) Sagittal left-sided off-midline T2-weighted. (**g**) Sagittal left-sided off-midline black bone sequence. (**h**) Coronal STIR. A 12-year-old female with acute lower back pain when playing with friends. MRI demonstrates bilateral bone marrow edema centered in pars interarticularis (arrows). Fracture lines are also seen bilaterally in the pars (arrowheads). No spondylolisthesis is present.

**Figure 20 children-10-01094-f020:**
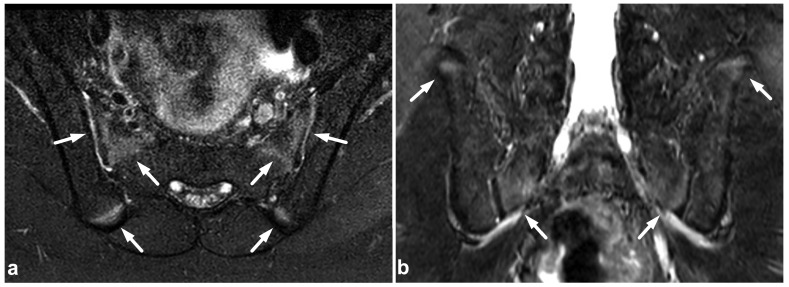
(**a**) Axial fat-suppressed T2-weighted. (**b**) Coronal STIR. An 11-year-old male, lower back/pelvic pain after falling on ice. MRI demonstrates symmetric zones of bright T2 signal (arrows) at the secondary ossification centers of the sacrum and iliac bone, characteristic of normal skeletal maturation. With the symptoms and injury mechanism in such a case, the findings could be misinterpreted as traumatic edema if the normal skeletal maturation and anatomy of the ossification centers are not kept in mind.

**Figure 21 children-10-01094-f021:**
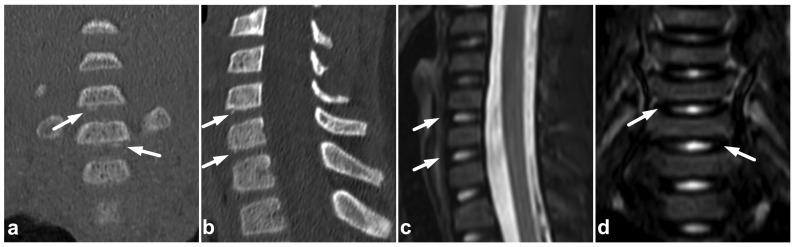
(**a**) NECT, coronal reconstruction with a bone kernel. (**b**) NECT, sagittal reconstruction with a bone kernel. (**c**) Sagittal STIR. (**d**) Coronal STIR. A 9-year-old male, neck pain and midline cervical tenderness after diving into shallow water. Small osseous fragments were seen at the anteroinferior corners of vertebral bodies C5 and C6 (arrows). The imaging appearance was consistent with inferior ring apophyses, but due to cervical spine symptoms and high-risk injury mechanism, MRI was performed. STIR imaging did not show any edema, neither in the proximity of the ring apophyses nor elsewhere in the spine. The apophyses were not visible at MRI.

**Figure 22 children-10-01094-f022:**
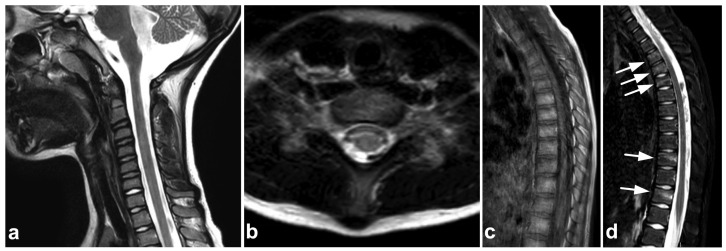
(**a**) Sagittal T2-weighted. (**b**) Axial T2-weighted. (**c**) Sagittal T1-weighted. (**d**) Sagittal STIR. A 6-year-old female after a car accident. The patient was referred for cervical and thoracic spine MRI due to a worrisome injury mechanism. The scanning was carried out without sedation or general anesthesia. The first sequence of the imaging protocol (**a**) has fine image quality without motion artifacts. The protocol’s fourth sequence (**b**) already demonstrates some motion artifacts, being yet sufficient for diagnostic purposes. The sixth sequence (**c**) suffers from distracting motion artifacts. After the sixth sequence, the patient was successfully encouraged to lie still for a few more minutes. The seventh sequence (**d**) was obtained, revealing bone contusions in the vertebrae Th3–5, Th10, and Th12 (arrows). In this case, the imaging protocol could have been further optimized; the sequences with presumably the highest sensitivity (STIR) should have been performed first.

**Figure 23 children-10-01094-f023:**
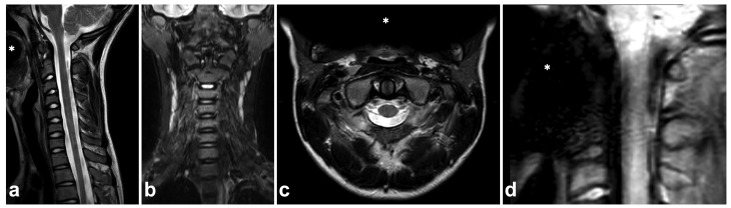
(**a**) Sagittal T2-weighted. (**b**) Coronal STIR. (**c**) Axial T2-weighted. (**d**) Sagittal PD-weighted, dedicated small-FOV upper cervical spine series. A 12-year-old male, flexion and axial load to the cervical spine in a trampoline accident. The patient wears braces. Metal-induced artifacts (asterisks) are seen, but they disturb assessments of the spine only in small-FOV PD-weighted imaging.

**Table 1 children-10-01094-t001:** Injury types, physiological findings, and imaging artifacts discussed in this review.

Bony Injuries	Ligamentous Injuries	Spinal Cord Injuries	Miscellaneous Injuries	Pitfalls
Bone bruises Simple compressions Burst fractures Chance fractures Avulsion fractures Posterior arch fractures	Craniocervical ligaments Posterior ligamentous complex Anterior and posterior longitudinal ligaments	Spinal cord transection Spinal cord contusion SCIWORA	Intervertebral disc injury Muscle injury Nerve plexus injury Vascular injury	Cervical pseudosubluxation Vertebral wedging Juvenile spondylosis Normal appearance of skeletal maturation Pulsation artifacts Motion artifacts Metal-induced artifacts

## Data Availability

Data availability does not apply to this article as no new data were created or analyzed in the study.

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
