# Peer review of "Emergency MRI in Spine Trauma of Children and Adolescents—A Pictorial Review"

_children, 2023, doi:10.3390/children10071094_

Round 1

Reviewer 1 Report

The authors prepared a pictorial review on the topic of MRI in pediatric spinal trauma. The topic is current in pediatric radiology and interesting for an average reader.

The review article is well prepared with great cases; therefore, I congratulate the authors for the well-done hard work!

I have only a few minor comments:

-          I think it would be important to mention SCIWORA already in the introduction part of the article, as this is one of the important specifics of pediatric spinal trauma and is often a reason why to consider doing MRI.

-          I would suggest (following introduction) to start the review by the paragraph “the suggested imaging protocols and practical considerations”, before going into the in-depth description of all pathologies. At the start of this paragraph, I would advise to first list the key indications where MRI is useful – state indications when it is necessary and recommended to perform MRI. Also state where it is not necessary, but it may be of added value. Are there any differences for children under 5 years of age when general anesthesia cannot be avoided?

-          In conclusion, please add that the need for sedation can be diminished, especially in children older than 5 years. Imaging of younger children would still require it.

The figure captions should be directly under the figures – part of the figure captions is inserted as regular text right now. Usually, sex of the patient is also presented in figure caption. Figure markings could also be improved – arrowheads and arrows look very similar, black arrows can sometimes be hardly seen. Some of the figures don’t have markings (i.e. Fig 2 d) and would benefit from them to make it easier for the readers who are not radiologists to recognize the pathology (Children journal has a wide area of readers).

The Quality of English language is very good overall.

Reviewer 2 Report

This article provides a comprehensive overview of emergency MRI examinations in pediatric spinal trauma. Although this is a rare disease group, diagnosis and treatment decisions must be made after prompt examination. In this regard, I believe that this review, which includes many case examples, will be of value to readers. The review also includes many points that are different from those in adult spinal trauma, but these points are also adequately mentioned. I believe that this review deserves to be accepted for this paper. Only the following issue need to be revised.

The following is described in the conclusion. “The need for sedation can be diminished with calm and child-friendly conditions and dedicated staff, making MRI more approachable in clinical practice.”  However, in real clinical practice, anesthesia and sedation are often necessary during MRI imaging in children, and it is better to describe what specific administration protocols are recommended, with a discussion of the literature.

Reviewer 3 Report

Thank you for inviting me to review this paper. The authors discussed the importance of MRI in peds spinal trauma through a pictorial review.  the only thing that I recommend is changing it to a case-series rather than a review since the authors have added patients information. it might still be exempt from IRB review. In that case, a small method section would be required. I would also add a table or a diagram that summarize the injuries discussed to make it easier for readers to have a quick look over the study. 
